# *Ferula communis* Root Extract: In Vitro Evaluation of the Potential Additive Effect with Chemotherapy Tamoxifen in Breast Cancer (MCF-7) Cells Part II

**DOI:** 10.3390/plants12051194

**Published:** 2023-03-06

**Authors:** Jessica Maiuolo, Natalizia Miceli, Federica Davì, Irene Bava, Luigi Tucci, Salvatore Ragusa, Maria Fernanda Taviano, Vincenzo Musolino, Micaela Gliozzi, Cristina Carresi, Roberta Macrì, Federica Scarano, Anna Rita Coppoletta, Antonio Cardamone, Carolina Muscoli, Ezio Bombardelli, Ernesto Palma, Vincenzo Mollace

**Affiliations:** 1Laboratoy of Pharmaceutical Biology, IRC-FSH Center, Department of Health Sciences, School of Pharmacy and Nutraceutical, Faculty of Pharmacy, University “Magna Græcia” of Catanzaro, 88100 Catanzaro, Italy; 2Department of Chemical, Biological, Pharmaceutical and Environmental Sciences, University of Messina, Viale F. Stagno d’Alcontres, 31, 98166 Messina, Italy; 3Foundation “Prof. Antonio Imbesi”, University of Messina, Piazza Pugliatti 1, 98122 Messina, Italy; 4IRC-FSH Center, Department of Health Sciences, Faculty of Pharmacy, University “Magna Græcia” of Catanzaro, 88100 Catanzaro, Italy; 5PLANTA/Research, Documentation and Training Center, 90121 Palermo, Italy; 6Faculty of Pharmacy, San Raffaele Telematic University, 00042 Rome, Italy

**Keywords:** *Ferula communis* L., *F. communis* extract (FER-E), DPPH test, reducing power assay, ORAC assay, MCF-7, breast cancer, tamoxifen

## Abstract

*Ferula* L., belonging to the *Apiaceae* family, is represented by about 170 species predominantly present in areas with a mild–warm–arid climate, including the Mediterranean region, North Africa and Central Asia. Numerous beneficial activities have been reported for this plant in traditional medicine, including antidiabetic, antimicrobial, antiproliferative, anti-dysentery, stomachache with diarrhea and cramps remedies. FER-E was obtained from the plant *F. communis,* and precisely from the root, collected in Sardinia, Italy. A total of 25 g of root was mixed with 125 g of acetone (ratio 1:5, room temperature). The solution was filtered, and the liquid fraction was subjected to high pressure liquid chromatographic separation (HPLC). In particular, 10 mg of dry root extract powder, from *F. communis,* was dissolved in 10.0 mL of methanol, filtered with a 0.2 µm PTFE filter and subjected to HPLC analysis. The net dry powder yield obtained was 2.2 g. In addition, to reduce the toxicity of FER-E, the component ferulenol was removed. High concentrations of FER-E have demonstrated a toxic effect against breast cancer, with a mechanism independent of the oxidative potential, which is absent in this extract. In fact, some in vitro tests were used and showed little or no oxidizing activity by the extract. In addition, we appreciated less damage on the respective healthy cell lines (breast), assuming that this extract could be used for its potential role against uncontrolled cancer growth. The results of this research have also shown that *F. communis* extract could be used together with tamoxifen, increasing its effectiveness, and reducing side effects. However, further confirmatory experiments should be carried out.

## 1. Introduction

The plant kingdom offers a disproportionate amount of extracts and components responsible for carrying out beneficial biological activities in various health districts; however, its knowledge is still insufficient, therefore the natural bioactive wealth is destined to increase [1]. The potential of plant extracts also depends on many variables, including the part of the plant used, the chosen season, the latitude, the composition of the soil, the extraction mechanism and the bioavailability, among others [2,3].

*Ferula* L., belonging to the *Apiaceae* family, is represented by about 170 species predominantly present in areas with a mild–warm–-arid climate, including the Mediterranean region, North Africa and Central Asia [4]. *Ferula communis* L. is a perennial plant, with dense roots, that reaches up to 3 m in height; it presents stem erect, glaucous-purplish, finely striated, and branchy at the top. The lower leaves are petiolate, 3-6-pinnatosette, green on both surfaces, while the upper leaves are progressively reduced to conspicuous sheathing bases only. The flowers are gathered in numerous umbels, more or less sessile, with yellow petals. The fruit is achene elliptical or oblong-elliptical, strongly compressed dorsally [5,6]. *Ferula communis* is also nomined “straight stem plant” because of the rigidity of its wood, used, since ancient times, to produce chairs, stools and rods [7]. Numerous beneficial applications have been reported for this plant in traditional medicine, for example, *F. communis* has traditionally been used as antidiabetic, antimicrobial, antiproliferative and for the treatment of dysentery. Moreover, the administration of mashed fresh kernel is useful for the treatment of stomachache and the administration of plant seeds improves cramps; it is also used as an antidote to poisonous bites. Furthermore, the traditional use of this species against various ailments has been documented in many regions around the world. In Saudi Arabia, the rhizomes of this plant, which is known as “*Al-kalakh*”, are used locally as a traditional remedy for the treatment of skin infections, while the roasted flower buds are used against fever and dysentery. In Morocco, *F. communis* has traditionally been used as a hypoglycemic medicinal plant but its use has been restricted due to its toxicity. In Italy (Sardinia), this plant was used as an antiseptic [8,9].

This species has different biological properties due to bioactive compounds contained in the roots, but which also twist in leaves and rhizomes. Among these, sesquiterpenes [10], sesquiterpene coumarins [11], sesquiterpene lactones [12] and sulfur containing compounds [13] have been identified and are responsible for anti-inflammatory, antiproliferative, cytotoxic, antimicrobial and anti-neoplastic activities [14,15,16,17].

In recent decades, an estrogenic-like property of *F. communis* has been shown, justified by the presence of a compound known as ferutinin and chemically recognized as an ester of a sesquiterpenic alcohol. Another known compound is ferulenol (prenylated coumarin), responsible for a toxic effect that causes strong anti-coagulant properties (ferulosis) and death in animals taking it with food. Ferutinin and ferulenol are the most represented compounds of the non-poisonous and poisonous chemotype of *F. communis,* respectively [18,19]. Our research team has previously shown that when MCF7 cells were treated with *F. communis* extract (FER-E), a double dose-dependent effect was found: ferutinin is responsible for both hyperproliferative activity, at low concentrations (0.1–0.8 μM), and toxic property at higher concentrations (1.6–50 μM) [20]. Differently, in this experimental article we evaluated the whole root extract of *F. communis* at different concentrations (5–2500 μg/mL).

This scientific work aims to know the effects of administering the selective estrogen receptor modulator Tamoxifen (TAM) together with FER-E. In particular, we evaluated whether the concomitant association of FER-E improves or worsens the effectiveness of the anti-tumor drug. Subsequently, we evaluated the oxidative potential of the plant extract, to understand if its effect was or was not related to the oxidant or antioxidant property, notoriously maintained by most natural compounds [21,22,23,24].

## 2. Materials and Methods

### 2.1. Plant Material and Extraction Procedure

FER-E was obtained from the plant *F. communis*, and, precisely, from the root that was collected in April 2022, at Macomer, a small town in the province of Nuoro (Sardinia, Italy). The extraction protocol [20] provides that 25 g of root are mixed with 125 g of acetone (ratio 1:5, room temperature). The resulting solution was left to stand for 60 min in order to improve the extraction process. At the end of the indicated time, the solution was filtered to separate the liquid fraction from the solid part, rich in residues and impurities, and the liquid fraction was subjected to high pression liquid chromatographic separation (HPLC). The data generated by this method provided qualitative and quantitative information about the extract [20]. From an experimental point of view, HPLC analysis was performed on a Perkin Elmer Flexar Module equipped with a photodiode-array (PDA) detector, a series 200 autosampler, a series 200 Peltier LC column oven, a series 200 LC pump, and an Agilent 4 µm C18 100A (250 4.6 mm) column. The control of the HPLC system and data collection was accomplished online by a computer equipped with Chromera software (version 3.4.0.5712). An amount of 10 mg dry powder of root extract, from *F. communis,* was dissolved in 10.0 mL of methanol, filtered with 0.2 µm PTFE filter, and subjected to HPLC analysis. A two-solvent gradient (0.88% trifluoroacetic acid/acetonitrile) was used for the elution with a flow of 1 mL/min keeping the column at 30 °C. The wavelength of the detector has been set to 256 nm. The net dry dust yield obtained was 2.2 g. In addition, in order to reduce the toxicity of FER-E, ferulenol has been removed, as demonstrated by the HPLC spectrum shown in [20].

### 2.2. Free Radical Scavenging Activity

The free radical scavenging activity of FER-E was measured through the 2,2-diphenyl-1-picrylhydrazyl (DPPH) method [25]. The extract was tested in the range of 0.0625–2 mg/mL, and butylated hydroxytoluene (BHT) was utilized as positive control. A volume of 0.5 mL of each sample was mixed with 3 mL of methanol DPPH solution (0.1/mM) and incubated for 20 min at room temperature in the dark. Then, the color change of the solutions was estimated by measuring absorbance with a spectrophotometer (UV-1601, Shimadzu) at the wavelength of 517 nm. Three independent experiments were carried out, and the results are reported as the mean radical scavenging activity percentage (%) ± standard deviation (SD) and mean 50 % inhibitory concentration (IC_50_) ± SD.

### 2.3. Reducing Power Assay

The reducing power of FER-E was estimated through the Fe^3+^-Fe^2+^ transformation assay, according to the method of Oyaizu [26]. The extract was tested in the range of 0.0625–2 mg/mL and a volume of 1 mL of each sample was mixed with 2.5  mL of phosphate buffer (0.2 M, pH 6.6) and 2.5  mL of 1% K_3_Fe(CN)_6_. The mixture was incubated at 50 °C for 20  min. The resulting solution was cooled rapidly, mixed with 2.5  ml of 10% trichloroacetic acid, and centrifuged at 3000 rpm for 10  min. Finally, the upper layer of the solution (2.5  mL) was mixed with 2.5  mL of distilled water and 0.5  mL of 0.1% FeCl_3_. After incubation for 10 min of at room temperature in the dark, the color change of the samples was estimated by measuring absorbance at 700 nm. Three independent experiments were carried out, and the results are expressed as the mean absorbance values ± SD and ascorbic acid equivalent (ASE/mL) ± SD.

### 2.4. Ferrous Ions (Fe ^2+^) Chelating Activity

The Fe^2+^ chelating activity of FER-E was estimated by the method of Decker et al. [27]. The extract was tested at different concentrations (0.0625–2 mg/mL), and ethylenediaminetetraacetic acid (EDTA) was used as reference standard. Briefly, an aliquot (1 mL) of each sample was added to a solution of 2 mM FeCl_2_ (0.05 mL) and MeOH (0.5 mL). The reaction was initiated by the addition of 5 mM ferrozine (0.1  mL). The mixture was shaken vigorously and left standing at room temperature for 10 min; the absorbance was then measured spectrophotometrically at 562 nm. The control contained FeCl_2_ and ferrozine, complex formation molecules. The assays were carried out in triplicate, and the results are expressed as the mean inhibition of the ferrozine–Fe^2+^ complex formation (%) ± SD and IC_50_ ± SD.

### 2.5. Oxygen Radical Absorbance Capacity (ORAC) Assay

ORAC is a method that measures the antioxidant activity of a sample by evaluating the transfer of a hydrogen atom. In particular, the fluorescence loss of fluorescein (used as probe) is measured over time. This fluorescence is due to the formation of peroxylic radicals, following the spontaneous degradation of 2,2′-azobis-2-methyl-propanimidamide, dihydrochloride (AAPH) which occurs at 37 °C. The peroxylic radical oxidizes the fluorescein causing the gradual loss of the fluorescence signal. Antioxidants suppress this reaction and inhibit the loss of signal. Compound 6-Hydroxy-2,5,7,8-tetramethylchroman-2-carboxylic acid (Trolox) is a water-soluble analogue of vitamin E that inhibits the decay of fluorescence in a dose-dependent manner, acting as a positive control. Therefore, using the ORAC test, the antioxidant activity of any unknown sample can be measured by comparing the fluorescence values generated by fluorescein with those obtained from the curves of the different Trolox concentrations. Experimentally, we performed the ORAC test as follows:(1)Addition of fluorescein solution (20 μM in PBS);(2)Addition of the different Trolox concentrations (6.25, 12.5, 25, 50, 100 μM) and the sample to be tested (FER-E, 2mg/mL);(3)Addition of the buffer PBS as a negative control;(4)Addition of the AAPH (31.7 μM in PBS) to all wells.

The fluorescence signal is measured for about one hour (1 reading per minute) following excitation and emission at 485 and 538 nanometers, respectively.

A regression equation was constructed by comparing the net area under the decay curve of the fluorescein and the Trolox concentration. The area under the curve was calculated with the following equation:i = 60
AUC = 1 + Σ _f1/f0_
i = 1

### 2.6. Measurement of In Vitro Reactive Oxygen Species

H_2_DCF-DA is a molecule that easily diffuses into cells and, through intracellular esterases, is cleaved into H_2_DCF following the loss of the acetate group. H_2_DCF remains trapped within cells and is oxidized by intracellular ROS to form the highly fluorescent DCF compound. Quantification of the DCF probe provides the content of the ROS in the cell. MCF-7 cells were plated in 96-well microplates at a density of 6 × 10^4^ and, the following day, were treated with FER-E for 24 h. Subsequently, the growth medium was replaced with a fresh phenol red-free medium containing H_2_DCF-DA (25 μM). After 30 min at 37 °C, the cells were washed twice to remove the extracellular H_2_DCFDA, centrifuged, re-suspended in PBS, exposed to H_2_O_2_ (100 μM, 30 min) and the fluorescence was evaluated by flow cytometric analysis using a FACS Accury laser flow cytometer (Becton Dickinson).

### 2.7. Cell Cultures and Treatments

The human breast cancer cell line (MCF-7), human cervical cancer cell (HeLa) and immortalized Human Mammary Epithelial Cells (HBL-100) were purchased by the American Type Culture Collection (20099 Sesto San Giovanni, Milan, Italy).

These cell lines were maintained in culture in Dulbecco’s modified Eagle’s medium (DMEM), enriched with 100 U/mL penicillin, 100 µg/mL streptomycin, 10% heat-inactivated fetal bovine serum (FBS), in a humidified 5% CO_2_ atmosphere at 37 °C. When the cells reached a 50% confluence, they were treated with FER-E (5–2500 µg/mL, dissolved in DMSO) for 24 h, or with TAM (0.25–20 µM, dissolved in DMSO) for 24 h, or pretreated with FER-E for 24 h and then exposed to TAM for 24 h.

### 2.8. Proliferation Assay and Cytotoxicity Study

Cell proliferation was assessed by colorimetric assay 3-(4,5-dimethylthiazol-2-yl)-2,5-diphenyltetrazolium bromide (MTT) and this test is based on the observation that live cells possess mitochondria with active enzymes capable of reducing MTT to a dark blue visible reaction product form. Therefore, a reduction in MTT, measured calorimetrically, provides information on the cellular metabolic activity and the viability. Cells were seeded into 96-well plates (8 × 10^3^ cells/well) and, on the next day, the growth medium was replaced with medium containing FER-E (5–2500 µg/mL), or TAM (0–20 µM), as above described. Then, the medium was replaced with phenol red-free medium containing MTT solution (0.5 mg/mL) and, after 4h incubation, 100 μL 10% SDS was added to each well to solubilize the formazan crystals. The optical density was measured at wavelengths of 540 and 690 nm by means of a spectrophotometer (X MARK Spectrophotometer Microplate Bio-Rad).

The determination of cell mortality was carried out using the Trypan blue exclusion test, which allows to calculate the number of dead cells present in a cell suspension. It is based on the principle that living cells possess intact cell membranes that exclude Trypan blue dye, while dead cells do not. In this test, a cell suspension is mixed with Trypan blue and then visually examined to determine whether the cells absorb or exclude the dye. A vital cell will have a clear cytoplasm while a non-viable cell will have a blue cytoplasm. Experimentally, after treating the cells as planned, 1 part of 0.4% trypan blue and 1 part cell suspension are mixed. Counting should be performed after about 1–2 min. A drop of the mixture is applied in a hemacytometer where unstained (viable) and stained (non-viable) cells are counted under a microscope. Mortality is calculated as the ratio of the number of dead cells/ the number of total cells × 100 [28].

### 2.9. Cell Lysis and Immunoblot Analysis

To obtain the total cell lysate, cell monolayers in 100 mm plates were washed with ice-cold PBS and lysed with a pre-heated (80 °C) lysis buffer containing 50 mM Tris-HCl (pH = 6.8), 2% SDS, a protease inhibitor mixture and immediately boiled for 2 min. Conversely, to obtain the isolation of mitochondria, it was necessary proceed as described: (1) To centrifuge the cells for 10′ to ~370 g; (2) Resuspend the pellet in NKM buffer (TrisHCl 1 mM, pH 7.4; NaCl 0.13 M, KCl 5 mM, MgCl_2_ 7.5 mM); (3) Transfer the homogenate to a new test tube containing equal volume of 2 M sucrose; (4) Centrifuge at 1200 g for 5′ and transfer the supernatant to another test tube; (5) Centrifuge at 7000 g for 10′: the pellet contains the mitochondria; (6) Resuspend pellet in suspension buffer mitochondria (TrisHCl 10 mM, pH 6.7; MgCl_2_ 0.15 mM; sucrose 0.25 mM; PMSF 1 mM; DTT 1 mM). In both cases, the protein concentration was determined using a DCA protein assay. After the addition of 0.05% bromophenol blue, 10% glycerol, and 2% β mercaptoethanol, the samples were boiled again and loaded into SDS-polyacrylamide gels (12%). Following electrophoresis, the polypeptides were transferred to nitrocellulose filters, blocked with TTBS/milk (TBS 1%, Tween 20, and non-fat dry milk 5%), and then the antibodies were used to reveal the respective antigens. Primary antibodies were incubated overnight at 4 °C followed by a horseradish peroxidase-conjugated secondary antibody for 1 h at room temperature. The blots were developed using the chemiluminescence procedure. The following primary antibodies were used: a mouse monoclonal antibody for Cytochrome C (Thermo Fisher, 7H8.2C12) at 1:1000 dilution; a rabbit polyclonal antibody for Bax (Santa Cruz Biotechnology, A12009) at 1:2000 dilution; a mouse monoclonal anti-actin antibody (Sigma Aldrich) at 1:5000 dilution. Horseradish peroxidase-conjugated goat anti-mouse/anti-rabbit antibody were used as the secondary antibodies at 1:10.000 dilution.

## 3. Results

### 3.1. Cytotoxicity Induced by FER-E

In order to know the effects of increasing concentrations of FER-E on MCF-7 cell line, a dose–response curve has been built following treatment with the extract for 24 h. As can be seen in Figure 1a, low concentrations of FER-E (5–40 μg/mL) do not generate toxicity phenomena and viability is comparable to control. Vice versa, the highest concentrations (80–2500 μg/mL) increase mortality gradually and significantly. This result is superimposable with one previously found and published [20]. Increased mortality, at increasing concentrations of FER-E, is accompanied by modulated expression of Cytochrome c, which gradually increases with the dose of FER-E, as shown in Figure 1b. Similarly, the expression of protein Bax, at the cytoplasmic level, is reduced with the increase in FER-E, while at the same time, we can appreciate a slight increase in the expression of its mitochondrial isoform. The correct loading of samples on the gel is guaranteed by the expression of actin. Finally, the quantification of the expression of these proteins is shown in Figure 1c.

In order to assess whether the cytotoxicity, by FER-E, was identical in all cell lines, we carried out the same mortality test on other cell lines: ovarian cells (HeLa), derived from human cervical cancer cell and immortalized Human Mammary Epithelial Cells HBL-100. Hela cells have always been used for cancer research, AIDS, the effects of radiation and toxic substances, gene mapping and countless other scientific activities [29]. Their use, in this manuscript, is justified by the need to study, in addition to MCF-7, another human tumor line, and compare the effects generated by FER-E on cancer cells. In contrast, the epithelial human normal breast cells HBL-100 was identified in vitro in the milk of an apparently healthy woman [30]. They are immortalized healthy cells that tend, over time, to evolve towards tumor transformation. In this manuscript, the cell line HBL-100 was treated with FER-E to assess if there were differences when compared to overt cancer cells MCF-7. A comparative study of the effects of FER-E on the three cell lines (MCF-7, Hela and HBL-100), displayed in Figure 2, evidences that the tumor lines MCF-7 and Hela are susceptible to high concentrations of FER-E (80–2500 μg/mL) in equal measure, while normal human breast cells HBL-100 appear to suffer less the toxicity of FER-E, showing significantly lower mortality rates.

### 3.2. Antioxidant Activity

Since natural compounds usually exert antioxidant properties, we tested whether or not, this effect was present in our extract. For this purpose, we have measured the antioxidant properties of FER-E directly on the powder using different in vitro assays. In particular, the possible mechanisms of antioxidant action of the phytocomplex components were determined in the concentration range 0.0625–2 mg/mL. Both in the DPPH test and in the reducing power assay, based on the hydrogen atom transfer (HAT) and electron transfer (ET) mechanisms, the activity of FER-E started from 0.75 mg/mL and 0.25 mg/mL, respectively, and was found to be low as compared to the reference standard BHT. These results are shown in Figure 3a and b. Indeed, the extract displayed about 13% DPPH scavenging activity at the highest tested concentration (IC_50_ > 2 mg/mL), being the activity of BHT approximately 95% (IC_50_ = 0.07 ± 0.01 mg/mL). In the reducing power assay, an absorbance of 0.13 ± 0.01 was achieved with the maximum concentration of FER-E (ASE/mL value cannot be calculated), whereas BHT showed an absorbance value of 2.40 ± 0.03 (ASE/mL = 0.89 ± 0.06). In the Fe^2+^ chelating activity assay, which evaluates the inhibiting effect on the Fe^2+^-ferrozine complex formation, no activity was highlighted for the extract in the range of concentrations tested. Finally, in ORAC assay, the maintenance of the fluorescence signal is indicated as the area under the curve (AUC) measured over time. The relative value of FER-E was obtained by comparing its AUC to the standard antioxidant curves one, generated by different concentrations of Trolox. The results showed that this sample has no antioxidant capacity, deviating greatly from the curves for different Trolox concentrations and being similar to the obtained curve with blank sample, totally losing the fluorescence already after 10′. The results of the ORAC assay, relative to FER-E, are reported in Figure 3c.

### 3.3. Measuring the Oxidative Effect of FER-E on MCF-7 Cells

First of all, we evaluated if FER-E alone, could cause oxidative reactions and formation of ROS on MCF-7 cells. Subsequently, in order to establish its potential antioxidant effect, we pretreated the cells with FER-E and then exposed them to hydrogen peroxide, used as a positive control. As can be seen in Figure 4a, no concentration of FER-E causes oxidative damage, highlighted from the percentages of fluorescence that overlap the control. Cell exposure to hydrogen peroxide results in a sharp shift of fluorescence to the right, indicating the production and presence of ROS. Finally, pretreatment with FER-E cannot reduce the ROS generated by H_2_O_2_, which remains unchanged, as evidenced by the fluorescence percentages similar hydrogen peroxide one. Therefore, the results, in accordance with the in vitro measurement of the antioxidant potential of FER-E, have demonstrated the absence of protection against ROS. Relative quantification is shown in Figure 4b.

### 3.4. Additive Effect of FER-E and TAM

Breast cancer, to date, is one of the most common forms of cancer affecting women and also the leading cause of death from cancer. TAM is still used today as a chemotherapy for the treatment of breast cancer sensitive to hormones, estrogen (ER^+^) and progestin (ER^+^). Since FER-E has demonstrated like-estrogenic effects, we decided to use co-treatment FER-E + TAM on MCF-7 cancer cells. We initially tested the effects produced by treating with TAM at different concentrations (0.25–20 μM) for 24h. As can be seen in Figure 5a, treatment with TAM produces concentration-dependent effects with a statistically significant reduction in viability. In particular, cell viability and mortality data, following treatment with different TAM concentrations, have been evaluated by MTT (Figure 5a) and Trypan blue exclusion assay (Figure 5b), respectively; both methods gave the same results. In Figure 5c, we observed the effects of co-treatment FER-E + TAM. Surprisingly, co-treatment shows an additive effect of TAM and FER-E: in fact, the co-treatment FER-E 40 μg/mL + TAM 1 μM can generate a global response similar to TAM 5 μM one, anticipating the effect that the chemotherapy drug alone carries out at a higher concentration. Similarly, FER-E (80 μg/mL) +TAM (1 μM) anticipates the effect induced by TAM 10 μM alone. These results, obtained using the MTT test, were also confirmed by the evaluation of cell cycle phases under the same experimental conditions (Figure 5d and e). FER-E 40 μg/mL highlights an identical cell cycle profile to the control, while concentrations 80, 160 and 320 μg/mL show increasing changes in the phases of the cell cycle, represented, respectively, by (a) a reduction in the G1 phase, (b) the appearance of the sub-G1 phase (compatible with cell death), (c) a further reduction in G1 and a significant increase in sub-G1. The increasing concentrations of TAM (1, 5 and 10 μM) cause the total change in cell cycle profile with the gradual increase in cell death represented in sub-G1. Co-treatment FER-E (40μg/mL) + TAM (1 μM), shows a cell cycle profile similar to TAM 5 μM and, simultaneously, co-treatment FER-E (80 μg/mL) + TAM (1 μM) shows a cell cycle profile similar to TAM 10 μM alone. Figure 5 is displayed below.

## 4. Discussion

In Italy, *Ferula communis* is particularly widespread in all regions, since the climatic conditions coincide with the needs of the plant [31,32]. However, in the Mediterranean area, with its warm and dry climate, the fields of *F. communis* dominate the landscape [33]. In fact, our extract (FER-E) was obtained from the root of this species, collected in Macomer, in the province of Nuoro, (Sardinia, Italy). Numerous studies in the literature have also shown that some components of FER-E are responsible, at specific concentrations, for the toxicity of *F. communis* extracts [34,35,36,37,38,39,40]. This characteristic of the partial toxicity of FER-E could explain the double effect on cell viability generated by the extract: presumably the low concentrations do not cause any toxicity, while the higher doses consist of greater amounts of toxic coumarins, even after removing ferulenol [41,42]. This double effect may be also justified by the permeability of biological membranes to cationic species, such as calcium and magnesium, which undergo dose-dependent changes on a part of the sesquiterpenes [17]. So, toxicity of FER-E could be due to its ability to mobilize the calcium ion and induce apoptosis by activation of caspase 3 [43,44,45]. In our experimental model, the toxicity of FER-E is accompanied by induction of the apoptotic process, demonstrated by increased expression of Cytochrome c and modulation of Bax protein (Figure 1). Cytochrome c exerts numerous cellular functions: it deals with the transfer of electrons in the respiratory chain, acts as a detoxifying agent eliminating ROS and is particularly important in the induction of cellular apoptosis. This latter function involves the release of the protein from the mitochondria and its accumulation in the cytosol, responsible for the activation of an enzymatic cascade that leads to apoptosis [46,47]. The protein family Bcl-2, comprising at least 18 members, controls and regulates the apoptotic mitochondrial pathway. These components are divided into two groups: (1) the antiapoptotic or pro-survival Bcl-2 proteins (such as Bcl-2, Bcl-w, Bcl-xl and Mcl1); (2) the pro-apoptotic effector proteins (Bax and Bak) [48]. In healthy cells, Bax has a predominantly cytoplasmic localization [49]; in contrast, under stressful conditions, Bax is activated, its translocation in the mitochondria occurs, leading to the release of proapoptotic factors, such as Cytochrome c [50]. Treatment with increasing concentrations of FER-E caused the reduction in cytosolic BAX expression and the simultaneous slight increase in mitochondrial level [51]. In addition, we appreciated an increase in the expression of Cytochrome c in the cytosol. This modulation of the protein Bax and Cytochrome c is compatible with the activation of the apoptotic process. The involvement of apoptosis is also demonstrated by the distribution of cell cycle phases in our experimental conditions. In fact, treatment with increasing concentrations of FER-E causes a change In the cycle profile with a gradual reduction in the G1 phase and the appearance of increased cell death. Since increasing concentrations of TAM alone also result in death-compatible cellular damage, we can conclude that co-treatment FER-E + TAM produces an additive effect. Tamoxifen is a drug that belongs to the class of non-steroidal antiestrogens, substances that can counteract the effects of the hormone estrogen. Since hormonal therapies act by interfering with the production or action of particular hormones, Tamoxifen is widely used for the treatment of breast cancer, (female and male) both after the first-instance surgery, both after any relapse [52]. The principle of action of TAM is complex and not yet well understood, but its main function remains that of antiestrogen. Many forms of breast cancer need the presence of specific sex hormones, such as estrogen, to grow. On the surface of cancer cells, there are proteins, called receptors, which are sensitive to the presence of sex hormones. Under normal circumstances, when sex hormones are in contact with receptors, the neoplastic cells are activated, cell division is facilitated with the consequent growth of the tumor mass. TAM, simulating the action of estrogens, binds to receptors, but is not able to activate neoplastic cells, their division and growth. Therefore, in the presence of TAM, estrogens have no chance of reaching neoplastic cells, which as a result grow more slowly or do not grow any more [53]. Prolonged treatment with TAM produces numerous side effects, which are more common in pre-menopausal patients. The most common side effects are nausea, indigestion, hot flashes, sweat, increased appetite, depression, fatigue, dizziness, headache, thrombosis, vision disorders and increased risk of endometrial cancer [54]. Since it is known, in scientific literature, the estrogenic-like activity of *F. communis* [20,34,35] we decided to investigate what relationship could exist in the co-administration of FER-E and TAM. For this reason, we first of all, built a curve at increasing concentrations of the selective estrogen receptor TAM on our experimental model. The results showed a concentration-dependent viability reduction (0.25–20 μM), comparable to the data available in the literature [55]. We chose to continue the research using the concentration of TAM 1 μM for 24 h, capable of causing about 35–40% mortality. Interestingly, when we co-treated the cells with FER-E + TAM, an additive effect was detected: in particular, FER-E 40 μg/mL + TAM 1 μM was able to anticipate the effect generated by the treatment with TAM 5 μM alone. Similarly, FER-E 80 μg/mL + TAM 1 μM produced an equal effect to the TAM 10 μM one. These results show that FER-E are able to emphasize the effects of TAM, offering the possibility of using lower concentrations of the chemotherapy drug in breast cancer: this may reduce the side effects of TAM. It may be interesting to reduce the toxicity of TAM by using this drug with FER-E. Finally, it is important to note that these additive effects have been found, without any involvement of an oxidative property. In fact, the results obtained from four tests carried out directly on the extract (DPPH, Reducing power assay, chelating capability on Ferrous ions and ORAC assay, Figure 3) and from one test carried out on the cell line MCF-7 (Figure 4), have shown that the effects of FER-E are independent of its oxidative properties. The ability of FER-E to induce cytotoxicity, at certain concentrations, appeared to be directed in particular to cancer cells. In fact, a comparative study, involving two cancer lines (MCF-7 and Hela) and a normal one (HBL-100) showed that healthy cells demonstrate a lower toxicity to the extract than cancerous (Figure 4). This result invites us to consider FER-E as a plant extract selectively able to possess, at high concentrations, an anticancer effect. However, this characteristic, to be considered true, requires numerous confirmations: in addition to other in vitro, in vivo results and clinical trials, it would be expected an unequivocal involvement of a pathway that is activated in cancer cells and simultaneously switched off in healthy cells.

## 5. Conclusions

In conclusion, it is possible to say that FER-E is able to create an additive effect with the chemotherapeutic TAM, with a mechanism different from the oxidative one. This co-treatment effect could ensure the use of reduced concentrations of TAM, maintaining the same effectiveness and reducing side effects. Furthermore, high concentrations of FER-E could also be used with an antiproliferative effect in the uncontrolled growth of MCF-7 cancer cells. In fact, our preliminary studies have shown that the effect of cytotoxicity is particularly expressed in cancer cell lines, rather than in healthy cells. These results are very interesting but need further investigation in order to confirm or disprove these data and not to consider them purely speculative.

## Figures and Tables

**Figure 1 plants-12-01194-f001:**
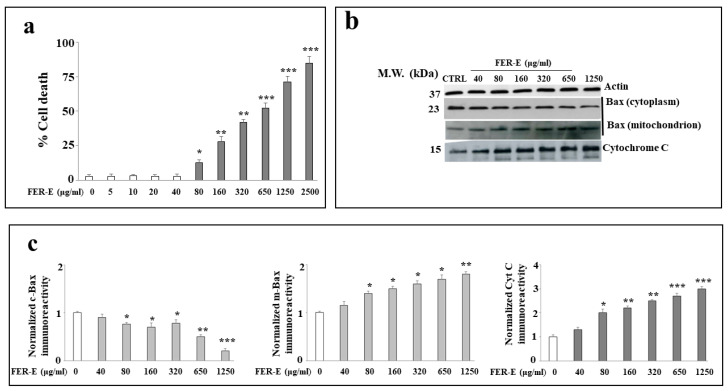
Effects of FER-E on cell viability. Exposure of FER-E to the cell line MCF-7 produces two effects: low concentrations (5–40 μg/mL) appear non-toxic, while high concentrations (80–2500 μg/mL) result in a gradual and significant increase in mortality, as shown in (**a**). In (**b**), the expression of proteins Cytochrome C and Bax (cytoplasmic and mitochondrial), both linked to the induction of apoptosis, was evaluated following treatment with increasing concentrations of FER-E, as shown in the figure. Results have been normalized thanks to the hausekeeping protein actin. In (**c**), the respective quantifications are shown. Three independent experiments were carried out, and the values were expressed as the mean ± standard deviation (sd). * denotes *p* < 0.05 vs. the control; ** denotes *p* < 0.01 vs. the control; *** denotes *p* < 0.001 vs. the control. Analysis of Variance (ANOVA) was followed by a Tukey–Kramer comparison test.

**Figure 2 plants-12-01194-f002:**
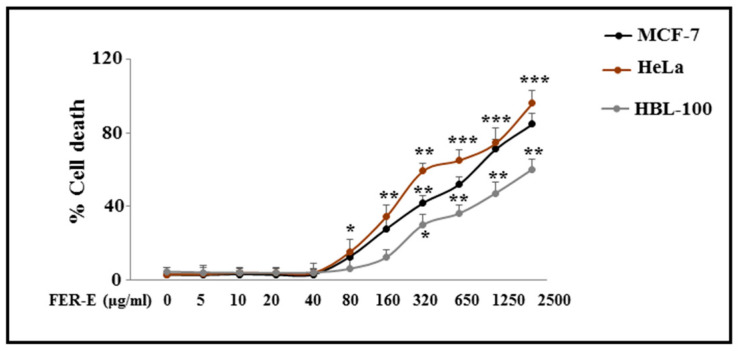
Comparison of FER-E induced cell mortality in different cell lines. The curves shown in the figure are related to mortality caused by treatment with FER-E (80–2500 μg/mL) at different concentrations, for 24 h, in three cell lines (MCF-7, Hela and HBL-100). Three independent experiments were carried out, and the values were expressed as the mean ± sd. * denotes *p* < 0.05 vs. the control; ** denotes *p* < 0.01 vs. the control; *** denotes *p* < 0.001 vs. the control. Analysis of Variance (ANOVA) was followed by a Tukey–Kramer comparison test.

**Figure 3 plants-12-01194-f003:**
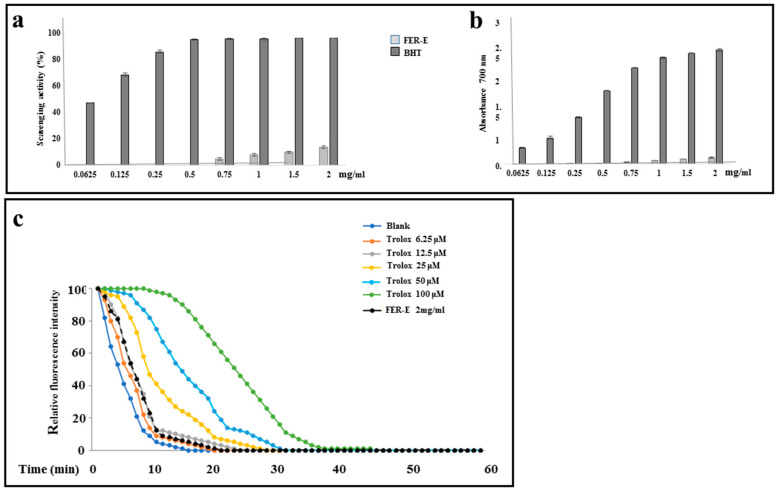
In (**a**,**b**) free radical scavenging activity (DPPH assay) and Reducing power of FER-E are reported, respectively. Reference standard: BHT (for both panels). In (**c**) the results obtained by the ORAC test are displayed, according to the manufacturer’s instructions in black 96-well plates. The fluorescence measurements, made at 485 nm excitation and 535 nm emission respectively, were carried out every 60 s, over a total measurement period of 60 min. AAPH was added into each well to start the generation of ROS. Initially, a standard antioxidant curve was created using different concentrations of Trolox (0–50 µm) in PBS. The antioxidant capacity, expressed as the area under curve (AUC), was calculated for each Trolox Concentration. Curves showing the loss of fluorescence were constructed for different concentrations of Trolox and, for completeness, a “blank sample” was also measured. Finally, our sample FER-E (2 mg/mL) was evaluated and compared to the other samples examined. This assay was carried out in triplicate and a representative experiment is shown.

**Figure 4 plants-12-01194-f004:**
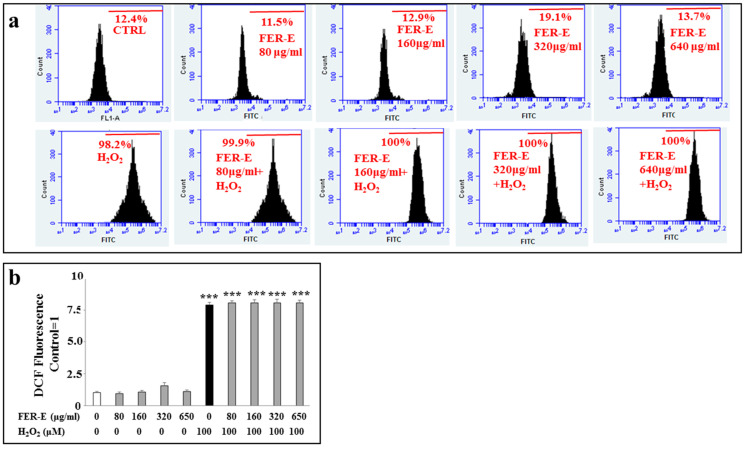
Measurement of the potential antioxidant activity in vitro of FER-E. In (**a**) the line colored in red, at the top of every plot, was drawn on the control and kept identical for all treatments. It identifies the percentage of cells that are included in the underlying area. The increase in ROS causes a shift on the right of the cell population. First of all, the oxidative capacity of FER-E (24 h) has been measured at different concentrations (80, 160, 320 and 650 μg/mL). Subsequently, the potential antioxidant property of this extract was evaluated and H_2_O_2_ (100 μM for 30 min) was used as a positive control. In (**b**), a quantitative representation of the obtained data is shown. The fluorescence percentage of the control was arbitrarily set = 1 and all other values were compared to it. Three independent experiments were carried out, and the values were expressed as the mean ± sd. *** denotes *p* < 0.001 vs. the control. Analysis of Variance (ANOVA) was followed by a Tukey–Kramer comparison test.

**Figure 5 plants-12-01194-f005:**
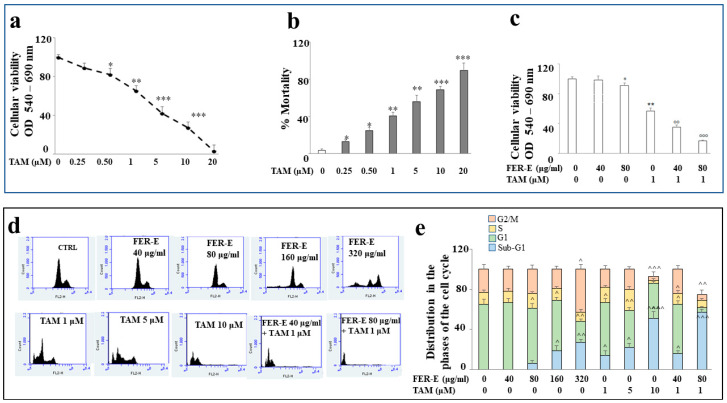
Additive effect of FER-E + TAM on MCF-7 cells. In (**a**) a dose–response curve of viability of MCF-7 cells, treated with increasing concentrations of TAM and measured through the MTT test, is shown. The spectrophotometric reading of the MTT test comprises values between 0 and 100, where 0 is considered to be the minimum viability and 100 the maximum. In (**b**) the same experiments were evaluated by Trypan blue exclusion assay mortality test. Both methods gave the same results. In (**c**) the results of the changes of viability, induced by the co-treatment FER-E + TAM, were highlighted. In this case, the MTT test was accompanied by cell cycle assessment (**d**), which confirmed the outcome of the experiment. In (**e**) the quantification of cellular cycle is highlighted. Three independent experiments were carried out, and the values are expressed as the mean ± sd. * denotes *p* < 0.05 vs. the control; ** denotes *p* < 0.01 vs. the control; *** denotes *p* < 0.001 vs. the control. °° denotes *p* < 0.01 vs. TAM 1 μM. °°° denotes *p* < 0.001 vs. TAM 1 μM. ^ denotes *p* < 0.05 vs. relative phase of the control. ^^ denotes *p* < 0.01 vs. relative phase of the control. ^^^ denotes *p* < 0.001 vs. relative phase of the control. Analysis of Variance (ANOVA) was followed by a Tukey–Kramer comparison test.

## Data Availability

The results are indicated in the paper. There is nothing else.

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
