# Peer review of "Ferula communis Root Extract: In Vitro Evaluation of the Potential Additive Effect with Chemotherapy Tamoxifen in Breast Cancer (MCF-7) Cells Part II"

_plants, 2023, doi:10.3390/plants12051194_

Round 1

Reviewer 1 Report

Review of the manuscript entitled “Ferula communis root extract: In vitro evaluation of the antioxidant properties and the potential synergistic effect with chemotherapy Tamoxifen in breast cancer (MCF-7) cells. Part II”.

In the manuscript submitted for review, the Authors described the study of antioxidant activity and the potential synergistic effect of FER-E and tamoxifen. Although the hypothesis has been well formulated and probably its verification could be an interesting contribution to the current state of knowledge, in the opinion of the reviewer, the way of presenting the results, their quality, discussion and minor shortcomings, such as linguistic and grammatical errors, inaccuracies in the presentation of data and methodological errors do not allow for acceptance of the manuscript in its present form. I therefore suggest rejection of the manuscript in its present form.

The manuscript might be reconsidered, but as a separate submission, after a thorough and detailed revision.

1. The authors devote a lot of space to the study of antioxidant activity. However, it is difficult to find the results of these analyzes at work. On the basis of the described methodology, we should find the results of at least three analyses, which cannot be found in the text. The results, even if negative, should be presented. Were the positive controls also ineffective, so maybe the test was poorly performed?

2. In order to rule out antioxidant activity, I suggest to run another test based on a cellular model, in addition to the ones used.

3. With such strong cytotoxicity, control and analogous studies using reference normal cells should be submitted.

4. Please show the native form of caspase 3, and reconsider to study other proteins that are involved possible cytotoxicity. In fact, the Authors do not indicate for what purpose they study c-casp3, moreover, in the text they write about casp-3, not c-casp3. Please note that c-casp3 levels are the same at any concentration above 1.6uM, while cytotoxicity (not cititoxicity) increases with concentration (between 1.6-12.8 almost 40% difference). This may suggest the involvement of other mechanisms in the observed effects.

5. Figure 2 is completely incomprehensible. It is not known what the authors presented here. Does the absorbance range from 0 to 100?? As for the viability/proliferation assay, it should be tested at one wavelength (possibly with a correction) and properly presented. If viability/proliferation is shown (the MTT test is used routinely in both studies, so it's hard to tell if we really have an effect on proliferation), how do we explain the difference for the concentration of 1.6 in graph 2.b and 1.6 in graph 1a. The use of another proliferation test could dispel these doubts (Please consider cell death and cell cycle analysis).

6. Many minor grammatical, stylistic and linguistic errors as well as mental abbreviations make the work difficult to read.

Author Response

Dear Reviewer,
thank you very much for your valuable advice. I hope that with these variations, the manuscript can be improved.

Reviewer 2 Report

Comments for Authors

The authors demonstrated that Ferula communis root extract increased the toxicity of the anticancer drug tamoxifen in breast cancer (MCF-7) cells. This result is remarkable, but the authors did not support their result by studying it on an animal model to confirm this result, and also, they did not study the mechanisms of this toxicity, even at the level of the cell that was studied. It makes the research incomplete and needs some complementary studies.

Major comments

1-The most important note here is that the authors calculated the concentrations that were used from a plant extract in μM, and this is not permissible and scientifically incorrect because the molecular weight of a mixture of several substances in the extract cannot be calculated you do not know the molecular weight (MW) of the plant extract that contains a mixture of active substances and not a pure substance that you can calculate its MW and its concentration in this way.

Concentrations are supposed to be calculated in milligrams per milliliter (mg/mL) in this research and the research that was previously published in the same journal.

2- Secondly, the authors did not study the synergistic effect mechanism of both tamoxifen and the plant extract, such as measuring the type of cell death and the concentration of reactive oxygen species in the cells used.

3-The title is very large and grandiose, despite the lack of results presented. Study of toxicity on one type of breast cancer cells for a period of 24 hours. The mechanisms of the plant's action with the anti-cancer substance were not established, which reduces the importance of the research and makes it incomplete.

specific comments

Title

1-The title is strange and does not agree with the results, because the author hid the results of the antioxidant and did not study the potential synergistic effect with chemotherapy. The study is just a study of the toxicity of the plant extract alone and in combination with tamoxifen in one type of cancer cells for a period of 24 hours.

Abstract

The authors mentioned that in addition the Reactive Oxygen Species (ROS) intracellular levels were determined in MCF-7 cells, this is not found in the research. Please clarify because this parameter is very important to prove the mechanism of action of the extract.

Methods

1-The method for the plant extraction is not clear, what is the method, is it in the presence of heat or not, and what is the period used to obtain the extract.

2-In the section on cell cultures and treatments, it should be mentioned how the plant extract and TAM were dissolved.

3-In the section on Proliferation assay and cytotoxicity study, 17- β-Estradiol (17-β-E2) should be omitted because it is not present in that study, and the concentration of Tamoxifen used here should be placed.

Results

1-The author mentioned three methods to estimate the antioxidant in the extract, but he did not mention a table indicating the results. Please put a table of the activity percentage, even if it is small.

2-In Figures 1 and 2, it is not permissible to calculate concentrations in μM, but it is correct to calculate them in mg / mL. Please repeat the figures with the correct concentration.

3- The title Measurement of in vitro reactive oxygen species should be deleted because that parameter was not studied.

Author Response

(The authors gave the same response as above.)

Reviewer 3 Report

In the study, authors aimed to examine antioxidant properties of extract of F. communis (FER-E) and synergistic potential with tamoxifen. Although the authors drew attention to the interesting effect of the studied extract in potentiating the cytotoxic effect of tamoxifen in the breast cancer tumor cell line, there are many aspects need to be considered and clarified, thus from my point of view, extensive revisions will be necessary.

Please consider the following points for the revision of your manuscript:

 1. In the manuscript, the characteristics of the tested extract are not stated. It is characterized only as extract of F. communis (FER-E) and the connection through the reference to its further characterization is only in method section as: In addition, in order to reduce the toxicity of FER-E, ferulenol has been removed, as demonstrated by the HPLC spectrum shown in [20]. Even in Abstract there is no specification of the composition of the extract or from which part of the plant it comes, or at least its type.

2. In Abstract and in Introduction, it is stated that: Since most plant extracts have a strong antioxidant activity, we tested whether this property was also exerted by the extract of F. communis (FER-E). (Abstract); First of all, since it is known that most natural compounds possess antioxidant properties… (page 3)

3. Please, provide references that would confirm this statement that most plant extracts exert a strong antioxidant effects.

4. Please, consider reformulation of that statement: Numerous beneficial activities have been reported for this plant in traditional medicine, including the anti-hysteria, anti-dysentery, expulsion of oral bloody humors, stomachache with diarrhea and cramps remedies. It could be questionable to report beneficial activity as anti-hysteria or expulsion of oral bloody humors.

5. In manuscript, it is stated that: The results of this research evidenced that high doses of FER-E could be used together with reduced concentrations of Tamoxifen, synergizing with this drug, increasing its effectiveness and reducing side effects. Since the results of that potential synergism are from one in vitro analysis only (and no in vivo experiments and clinical trials were done), it is not possible stated that this combination will reduce side effects.

Similarly, it is stated here: In fact, this dose may prove useful as the damage caused by TAM is not yet so serious that it cannot be repaired, mimicking the side effects of cancer patients taking this drug. In vitro conditions can never mimic complexity of human body. Then, in the results section, it was not proved that the damage of cells caused by that concentration of tamoxifen is reversible.

6. The biggest sticking point od the result section is in the chapter 3.2 Cytotoxicity induced by FER-E. Up to my knowledge, MCF-7 cells do not express caspase 3 (e.g., Reiner U. Jänicke. MCF-7 breast carcinoma cells do not express caspase-3. Breast Cancer Research and Treatment, 2008, 117 (1), pp.219-221. ï¿¿10.1007/s10549-008-0217-9). Please, provide explanation of that experiment showing expression of caspase-3 in MCF-7 cells.

7. In general, result section is very brief and, at least, half of the results are not shown, even they are negative.

8. Please, provide explanation, whether in Figure 1 (a) was perfomed and repeated the same analysis as in the article: Maiuolo, J.; Musolino, V.; Guarnieri, L.; Macrì, R.; Coppoletta, A.R.; Cardamone, A.; Serra, M.; Gliozzi, M.; Bava, I.; Lupia, C.; Tucci, L.; Bombardelli, E.; Mollace, V. Ferula communis L. (Apiaceae) Root Acetone-Water Extract: Phytochemical Analysis, Cytotoxicity and In Vitro Evaluation of Estrogenic Properties. Plants (Basel). 2022, 11(15), 1905.

9. In Discussion, it is stated that tested FER-E is comprising many other components. Please provide explanation how the molar concentrations in units of μM were achieved, since the antioxidant activity was reported to be tested in mass concentrations units of mg/mL.

10. The name of the chapter 3.4 Properties carried out by toxic and higher concentrations of FER-E does not correspond with the contents of this chapter as the results describe Synergistic effect of FER-E + TAM on MCF-7 cells and concentrations of FER-E used in that experiment were shown to be either non-toxic or slightly toxic.

11. Please, revise the sentence: These results show that toxic concentrations of FER-E are able to suppress the effects of TAM.

12. Please, correct form of word Citotoxicity”. Then, in cases as 0.1–0.8 μM ”, it is more appropriate to use dash instead of hyphen.

13. The specification of the incubation time is missing in the results of Figure 2b.

Author Response

(The authors gave the same response as above.)

Round 2

Reviewer 1 Report

I have read the revised version of manuscript. The Authors answered the most of requested problems pointed in my revision.  There are still some minor mistakes, but I hope the other Reviewers will monitor them. Thus I belive I can change my overall evaluation to minor revision. 

My only point, after a cursory review of the text, is to pay attention to the abbreviations and their correct explanation.

Author Response

Dear Reviewer,

we checked the manuscript according to your valuable suggestions.

Thank uou. Regards,

Jessica Maiuolo

Reviewer 2 Report

The authors have improved the manuscript but more corrections are needed before it can be approved for publication.

Specific comments

Title

1-The title needs to be changed to contain the most important results reached:

It doesn't make sense for the antioxidant to appear in the title because its results were negative. Also, just measuring the effectiveness of the extract with the anticancer drug tamoxifen in breast cancer (MCF-7) cells is not considered a synergistic effect.

The extract may increase the effectiveness of the drug in different ways, you need to study those mechanisms.

Introduction

1-The aim and objective of the study needs clarification.

Results

1-The authors reserve the right to report their negative results, but in a table, not a figure. Figure 1 is not useful at all and needs to be converted into a table, as I mentioned earlier.

2-The results contain 6 figures, not 5 as shown in the results. Please review carefully.

3-the arrangement of the figures is illogical, for example figure 3. (Comparison of FER-E induced cell mortality in different cell lines) ,It is supposed that Figure 4, needs to start with it the results and to be figure 1 and then put the results of effects of FER-E on cell viability of MCF-7 as figure 2.

4-The last figure of the Synergistic effect of FER-E + TAM on MCF-7 cells is not clear and needs to put the changes that occurred in the phases of the cell cycle in the form of columns and perform a statistical analysis.

5-The plant extract alone has an effect on programmed death (apoptosis) , as is evident from the results of the expression of  proteins cytochrome C and Bax, and it is also clear that TAM has the same effect from the results of the increase of cell death represented in G0 (apoptosis), so the combined effect of the FER-E + TAM  is additive and not Synergistic effect. Please rewrite that part based on the results of programmed cell death.

Discussion

1-I am surprised that the authors insist that the antioxidant activity of FER-E is important to explain the effect of the extract on cancer cell death, which is not true. The role of per-oxident measurement is much more important than the antioxidants. Free radicals are what cause cell death.

2-As it is clear from the results, the antioxidants, especially in the acetone extract, are not present, but other active substances may lead to cell death by activating caspases, for example, or the death of mitochondria. Unfortunately, the authors did not confirm these mechanisms.

3-The antitumor effects of tamoxifen are primarily attributable to modulation of gene expression via competitive inhibition of ER, which leads to inhibition of proliferation and an increase of apoptosis of breast cancer cells ( Wärri et al., 1993, Watts et al., 1994, Salami and Karami-Tehrani, 2003). The plant extract had the same effect alone or with tamoxifen, and there is no evidence of a different mechanism that makes the authors assume a synergistic effect. Please adjust the discussion based on the facts and findings of the authors

References

1-Wärri, A.M., Huovinen, R.L., Laine, A.M., Martikainen, P.M. and Härkönen, P.L., 1993. Apoptosis in toremifene-induced growth inhibition of human breast cancer cells in vivo and in vitro. JNCI: Journal of the National Cancer Institute, 85(17), pp.1412-1418.

2-Watts, C.K., Sweeney, K.J., Warlters, A., Musgrove, E.A. and Sutherland, R.L., 1994. Antiestrogen regulation of cell cycle progression and cyclin D1 gene expression in MCF-7 human breast cancer cells. Breast cancer research and treatment, 31, pp.95-105.

3-Salami, S. and Karami-Tehrani, F., 2003. Biochemical studies of apoptosis induced by tamoxifen in estrogen receptor positive and negative breast cancer cell lines. Clinical biochemistry, 36(4), pp.247-253.

Author Response

(The authors gave the same response as above.)
